# Facial Movements Extracted from Video for the Kinematic Classification of Speech

**DOI:** 10.3390/s24227235

**Published:** 2024-11-12

**Authors:** Richard Palmer, Roslyn Ward, Petra Helmholz, Geoffrey R. Strauss, Paul Davey, Neville Hennessey, Linda Orton, Aravind Namasivayam

**Affiliations:** 1School of Earth and Planetary Sciences, Curtin University, Perth, WA 6102, Australia; petra.helmholz@curtin.edu.au; 2School of Allied Health, Curtin University, Perth, WA 6102, Australia; r.ward@curtin.edu.au (R.W.); g.strauss@curtin.edu.au (G.R.S.); p.davey@curtin.edu.au (P.D.); n.hennessey@curtin.edu.au (N.H.); linda.orton@postgrad.curtin.edu.au (L.O.); 3Department of Speech-Language Pathology, University of Toronto, Toronto, ON M5G 1V7, Canada; a.namasivayam@utoronto.ca

**Keywords:** digital biomarkers, kinematics, spatiotemporal profiling, Speech Sound Disorders

## Abstract

Speech Sound Disorders (SSDs) are prevalent communication problems in children that pose significant barriers to academic success and social participation. Accurate diagnosis is key to mitigating life-long impacts. We are developing a novel software solution—the Speech Movement and Acoustic Analysis Tracking (SMAAT) system to facilitate rapid and objective assessment of motor speech control issues underlying SSD. This study evaluates the feasibility of using automatically extracted three-dimensional (3D) facial measurements from single two-dimensional (2D) front-facing video cameras for classifying speech movements. Videos were recorded of 51 adults and 77 children between 3 and 4 years of age (all typically developed for age) saying 20 words from the mandibular and labial-facial levels of the Motor-Speech Hierarchy Probe Wordlist (MSH-PW). Measurements around the jaw and lips were automatically extracted from the 2D video frames using a state-of-the-art facial mesh detection and tracking algorithm, and each individual measurement was tested in a Leave-One-Out Cross-Validation (LOOCV) framework for its word classification performance. Statistics were evaluated at the α=0.05 significance level and several measurements were found to exhibit significant classification performance in both the adult and child cohorts. Importantly, measurements of depth indirectly inferred from the 2D video frames were among those found to be significant. The significant measurements were shown to match expectations of facial movements across the 20 words, demonstrating their potential applicability in supporting clinical evaluations of speech production.

## 1. Introduction

Speech-Language Pathologists (S-LPs) are trained in the assessment and management of Speech Sound Disorders (SSDs) in children. The clinical evaluation of SSDs typically involves informal assessments and the administering of standardized tests [1] which entail the child saying a list of sounds in words and phrases that have been selected to elicit jaw, lip, and tongue motor movements known to be involved in the correct production of these sounds [2]. Through expert perceptual analysis of a child’s speech patterns, errors in sound production, and observation of the associated orofacial movements, an S-LP can determine if an SSD is likely to be present. Years of specialist training and experience are needed to reliably identify disordered speech production [3,4].

Accurate and timely differential diagnosis followed by appropriate therapeutic intervention mitigates the long-term negative consequences of SSDs in children [5,6]. Identifying atypical speech patterns sufficiently early in development allows for interventions to have the desired effect of preventing errors in speech production from becoming entrenched and potentially leading to hard-to-treat persistent speech errors [7]. Unfortunately, there are significant barriers to SSD diagnosis in children which may include a range of issues from lengthy and variable assessment practices, to differences in clinical expertise and experience, to inefficient low-technology assessment practices, with diagnostic accuracy compromised by insufficient time to complete the detailed level of analysis required for the diagnosis of an SSD [3,4,8,9]. Due to the perceptual nature of the assessments, evaluation is also subjective and there may be large discrepancies between different S-LPs that can contribute to the analysis of an individual child’s error patterns leading to poor reliability even between those considered to be experts in the field [3,4,9,10]. Given these issues and the time required to complete a thorough assessment, there is great interest and cost-benefit motivation to augment speech examination with automated objective methods to improve the speed, reliability, and translatability of such assessments [3,4,9,10].

In recent years, researchers have investigated computer vision-based methods of measuring facial movements to help mitigate assessor bias and begin to move toward more objective and standardized criteria for identifying atypical speech patterns [11,12]. Such approaches have shown promise in classifying atypical speech movements that can signify the presence of specific diseases [12,13,14]. However, these approaches have typically used only a small number of features and employ high-level and low-frequency summary statistics such as range-of-movement, or maximum velocity in a single orientation over the duration of a word or sound’s production [15]. This limits these approaches in their ability to characterize facial movement in sufficient detail to allow a thorough study of how variation in facial movement relates to differences at the level of clinical assessment.

We have developed a system that uses a state-of-the-art facial mesh detection and tracking algorithm to automatically extract and record expertly identified measurements of clinically salient facial movements from high-definition video recorded at 60 frames per second [16]. In this study, we use this system to analyze video recordings of 51 adults and 77 children (in two different age bands) with typical speech motor control saying 20 words taken from the mandibular and labial-facial stages of the Motor Speech Hierarchy based Probe Words (MSH-PW) list [2]. We evaluate each of the facial measurements extracted from the recordings for its ability to classify the words correctly and show which measurements exhibit classification performance at the α=0.05 significance level.

### 1.1. Hypotheses

The facial mesh detector used in this study is able to estimate the depth of a point on the face from a single two-dimensional (2D) video frame. The ability to measure posterior/anterior movement in the jaw and/or mouth and lips, and whether such movement is typical has important clinical implications for the objective assessment of SSDs. In this work, we evaluate several automatically extracted measurements of depth for their efficacy in the word classification task as a first step in justifying whether further investigation is warranted in tasks more closely aligned with clinical use. Our first hypothesis is that measurements of facial movement in depth will show significant efficacy in word classification.

Evaluating the extracted measurements against both typically developed adults and typically developing children allows for a comparison of classification performance between the two cohorts. Whilst speech articulation differences between adults and very young children are known to exist [17], due to the slightly older age of the child participants in this study our second hypothesis is that the facial measurements having the best classification performance will be the same across both the adult and child cohorts.

The words in the mandibular and labial-facial stages of the MSH-PW list are selected so as to elicit specific motor actions [2]. Our third hypothesis is, therefore, that the words sharing similar facial movements will best be classified by the measurements designed to capture those expected movements.

In this work, a wide variety of facial measurements are evaluated including measurements of lateral deviation from the mid-sagittal plane as well as lateral asymmetry in the superior/inferior and anterior/posterior directions. Typical speech production should not involve asymmetric facial movements in any axis [18]. Given that the participants are typical in their speech production abilities for their age, we expect no asymmetric jaw or mouth movements in the production of the words. Thus, our fourth hypothesis is that measurements of asymmetric movement will not perform significantly well in the word classification task.

The automatically extracted facial measurement distances are normalized by a calculation of the participant’s face size. This is conducted to facilitate the comparison of the measurements between participants. The facial size measurement is calculated in every frame and is designed to vary as minimally as possible while the participant speaks. Because it is incorporated as a normalizing factor within the majority of the extracted distance measurements it is important to show that this measurement alone is not capturing any salient information about facial movements. As such, our fifth and final hypothesis is that the “Face Size” measurement will not by itself perform significantly well in the word classification task.

### 1.2. Significance

This work is unique and significant for several reasons. Firstly, it shows that a state-of-the-art facial mesh detection and tracking algorithm that can infer depth directly from two-dimensional (2D) video in near real-time can extract measurements of facial movements that are sufficiently accurate to distinguish between clinically salient speech movements. Capturing facial movements in depth from single stream 2D video is a relatively recent technological development and showing its potential capability for clinical applications is extremely valuable due to the ubiquity of low-cost 2D cameras.

Secondly, our selection of facial measurements is guided by clinical expertise and an understanding of how the measurements capture facial motions of interest. From a large set of these features, we empirically find those that perform best. This is unlike most other approaches which either use a minimal set of more conveniently measured features or take an undirected approach to learning the features that work best for a given classification task but can result in features being used which have little relation to existing clinical understanding.

Thirdly, in our approach, the differences in facial movements are determined over the entire duration of each word’s production with individual measurements extracted at 60 frames per second (equivalent to the frame rate of the recorded video). This allows for a much richer investigation of changes in facial motion as they relate to speech production.

Finally, we have built the current system for eventual real-world use as a fully realized software application that allows statistical kinematic profiles from different demographic groups to be investigated. The system is extensible across multiple forms of speech assessment that may require different words to be spoken by patient populations.

## 2. Materials and Methods

### 2.1. Participants

Three participant cohorts were defined: adults, children between the ages of 3 to 3.5 years, and children between the ages of 3.5 to 4 years. Fifty-one adult participants (27 male, 24 female) between 18 and 65 years of age (uniformly distributed), 36 children in the younger age band (21 male, 15 female), and 41 children (18 male, 23 female) in the older age band were recruited from the Perth metropolitan region of Australia. The average age for children in the younger age band was 38.14 months (range 36 months and six days to 41 months and 27 days) and 44.73 months for those in the older age band (range 42 months and 4 days to 47 months and 29 days). The child participants were assessed for typical development of their speech and language skills using the Clinical Evaluation of Language Fundamentals, Australian and New Zealand Preschool, 2nd Edition (CELF-P2), and the Goldman-Fristoe Test of Articulation 3rd Edition (GFTA-3) Sounds-in-Words Subtest as shown in Table 1.

All participants were fluent speakers of English and represented a cross-section of English dialects and accents (e.g., Australian, English, American, Canadian, European). Hearing was screened at 1, 2, 4 and 8 kHz. Age-related hearing changes were detected in participants older than 58, without any noticeable impact on speech production.

### 2.2. Stimuli

Twenty words based on developmental speech motor complexity were used in the study [2]. Ten words targeted mandibular opening and closing movements (e.g., “Map”, “Bob”, “Pam”) and 10 words targeted labial-facial rounding and retraction movements (e.g., “Bush”, “Feet”, “Phone”). These words had been validated in an earlier study for assessing speech motor skills in children [2]. The scoring of these words is based on qualitative descriptions provided by S-LPs. S-LPs are typically asked to focus their attention on the specific orofacial movements characteristic of each word and provide a binary rating. For example, whether appropriate mandibular range and control (no anterior or lateral slide) or appropriate lip symmetry and rounding/retraction is present during the word’s production. Unusual motor control in these characteristic movements may indicate some form of atypical speech development [19,20].

### 2.3. Procedures

In a quiet lab space, participants were seated in a comfortable chair and asked to repeat the 20 words displayed on a digital screen, following the prompt “say X”. Before starting the task, child participants engaged in free play activities, such as reading picture books, playing with dolls, and building with construction materials to familiarize them with the laboratory environment. The adult participants were instructed to speak naturally with no unusual emphasis, accentuation, pause, or inflection. The child participants were provided with a repeat opportunity to say the word if the assessor did not have the child’s attention or their face was obstructed from the camera’s view. Common obstructions included hand(s) in front of the face, a bowed head, or head turned away. The best production was retained.

The video was captured from a front-facing camera positioned approximately at the participant’s eye level. The camera used was a Blackmagic Pocket Cinema Camera 4K with an Olympus M.ZUIKO Digital ED 45 mm F/1.8 lens sourced from Camera Warehouse, Australia. Video was recorded at 60 frames per second with HD (1080 p) resolution output. The camera was placed at a distance from the participant to allow their face to fill half the frame height. The start and finish (onset and offset) times of each word in each video were manually identified by a single S-LP with 25+ years of clinical experience in motor speech disorder assessments to ensure consistency in identification. Due to differing vocalizations, the criteria for identifying onset and offset times differed for each word. Table A1 in the Appendix A details the per-word onset and offset criteria used.

#### 2.3.1. Extraction of Facial Measurements

The qualitative descriptions of mandibular and labial-facial movements associated with the typical spoken production of the 20 words were translated by an S-LP into a set of 29 specific objective measurements of jaw and lip movement in three spatial dimensions. The measurements were chosen because they were determined to characterize either typical facial movements in the production of the words, or facial movements able to discriminate between a given word and others in the set, or a given word and its atypical production. These base measurements along with their first and second-order derivatives (velocity and acceleration) were extracted from video recordings of the participants speaking the target words.

The video from each participant’s production of the 20 words was passed through custom software to automate the extraction of measurements from each video frame within the duration of each word. Half a second of padding before/after the onset/offset defined by the expert S-LP was used to ensure these timings could be refined. The Blazeface detector [16] which is able to infer the 3-Dimensional (3D) spatial positions of 478 facial points from a single 2D image or frame of video was used as the underlying facial mesh detector. A subset of these points was used to localize the positions of standard anthropometric clinical facial landmarks around the jaw and mouth [21]. An explanation of how the Blazeface detector maps 2D image pixels to 3D spatial points is outside the scope of this work and the reader is referred to the original source for details [16]. Table 2 shows the association between the numbered points of the facial mesh output by the Blazeface detector [16], the standard clinical facial landmarks, and the extracted measurements (as described in more detail in Table 3).

When deriving the positions of the clinical facial landmarks from the Blazeface facial mesh points, the closest (or pair of closest) points were used. The positioning on the face of the Blazeface facial mesh identifiers can be seen in high resolution at https://storage.googleapis.com/mediapipe-assets/documentation/mediapipe_face_landmark_fullsize.png (accessed on 15 October 2024).

Prior to extraction of the measurements, head orientation was detected and corrected using landmarks from the upper face and temple because of their stability relative to motion in the lower face. The Blazeface detector was able to infer positions of facial landmarks in the case of self-occluding head rotation angles though the precision of point estimation was affected. Figure 1 shows the detection of the salient facial landmarks and the head orientation from a single frame of video of a face captured using an HD Webcam. Red and blue dots indicate the left and right bilateral landmarks, respectively, whilst landmarks on the midline of the face are shown in white. The red, green, and blue lines depict the X, Y, and Z facial axes, respectively. The main facial features are outlined in yellow for reference.

Variation in each of the distance measurements was normalized using a standard measurement of the participant’s “Face Size” derived empirically from a small sample of adult participants incorporating a weighted combination of different distance measurements around the forehead and eyes. This normalization factor was designed to vary as minimally as possible while the participants spoke. Normalization was performed to facilitate the comparison of measurements across participants having different apparent facial dimensions. Figure 2a shows how the “Face Size” measurement varies minimally compared to another of the measurements (Mouth Opening/“Labial Fissure Rounding”) shown in Figure 2b for participants in the adult cohort saying the word “Pam”.

Inter-landmark distance measurements were decomposed into the three directions of movement to separate out motion along the lateral (X) axis from motion in the superior/inferior (Y) axis from motion in the anterior/posterior (Z) axis. This was achieved by projecting the 3D measurement vector along the component orientation vectors of the participant’s head. For each participant, the facial coordinate origin was set as the midway point between the two bilateral tragion landmarks and displacements were measured with respect to this point. The measurements, the expected character of movement they were designed to capture, and the associated facial landmarks involved are shown in Table 3. Measurements suffixed with ^+^ are scalars. All others are a composition of three separate values corresponding to the three independent spatial directions (left/right lateral, superior/inferior, anterior/posterior). All measurements except those suffixed with an asterisk are normalized using the aforementioned “Face Size” factor.

Bilinear interpolation was used to fit splines to the data points from each measurement to produce spatiotemporal charts of change in each measurement over the duration of each word’s production by a participant. The onset and offset times defined by the expert S-LP were then automatically adjusted against one of the extracted measurements—“Mouth Opening”—using an algorithm to minimize the residual sum of square differences across each participant cohort for each word. Each participant cohort was aligned independently of the other two. Differences in individual speaking rates meant that spoken word durations were linearly stretched or compressed to achieve alignment with others in the cohort. Such linear time normalization techniques have been successfully applied to speech kinematic data previously [22]. Note that variation of speaking rate within the production of a single word was not accounted for in the linear time normalization of spoken word durations. The automatically adjusted timings were then reappraised by the S-LP to confirm the final timing alignment of every word across each cohort. This step was carried out both to provide a way of confirming the manually decided timing alignment, but also to help establish and clarify the objective criteria used in the definition of each word’s onset and offset.

#### 2.3.2. Validation

A leave-one-out cross-validation procedure was chosen both to maximize the number of observations per sample as well as the amount of data available for training each classification model. Validation was carried out on each participant cohort separately. In this approach, each participant in the cohort was, in turn, set as the test participant, with the remaining participants used to train the classifier. For each word/measurement pair in the training set, the data points were interpolated in duration as previously described and averaged at 1000 equally spaced intervals. This resulted in spatiotemporal profiles for each word/measurement pair showing a change in the measurement’s value (averaged across the participant cohort) together with standard deviation. A similar interpolation of the test participant’s word/measurement pairs was performed. Figure 3 shows an example spatiotemporal profile of the “Lip Action (Y)” measurement generated from all 51 participants in the adult cohort saying the word “Map”.

For each measurement and its first and second-order derivatives (velocity and acceleration), the similarity of the chart produced from each word spoken by the participant under test was compared against that of each word in the average profile created by the remaining participants in the cohort. The metric of similarity ψ on measurement *m* and word *w* was calculated as the average squared difference of each participant’s measurement *t* from the corresponding profile *p* of that measurement over 1000 interpolation points:$$\psi \left(mw\right)=\frac{1}{1000}\sum_{i=1}^{1000}{\left(p{\left(mw\right)}_{i}-t{\left(mw\right)}_{i}\right)}^{2}.$$

The predicted word was selected as the one having the minimum corresponding ψ over the 20 word set *W*:$$prediction_{m}=\underset{w\in W}{arg min}\psi \left(mw\right).$$

This procedure was conducted for each word across the participant cohort and the classification predictions recorded in a 20 × 20 confusion matrix for each measurement resulting in a sample size of 20 × *N* observations (for *N* participants in the cohort) per measurement.

The 20 × 20 confusion matrices were also aggregated into 5 × 5 matrices based on the similarity of facial movements across the word set resulting in three classes in the mandibular group, and two in the labial-facial group. The obvious distinguishing characteristics of the words in each class are shown in Table 4. Note that for Lip Shape, the vertical jaw height is based on vowels [23]. “Papa” was excluded from aggregation at the similar movement level due to the characteristics of its kinematic production being dissimilar to all of the other words. The classification results were also aggregated into 2 × 2 matrices to evaluate classification performance between the mandibular and labial-facial groups (“Papa” was reincluded here).

For each resulting matrix, true positives ($T_{P}$), false positives ($F_{P}$), and false negatives ($F_{N}$) were counted and Recall (*R*) and Precision (*P*) calculated in standard fashion as: $$\begin{array}{ccc} R & = & T_{P}/\left(T_{P}+F_{N}\right),\\ P & = & T_{P}/\left(T_{P}+F_{P}\right). \end{array}$$

To represent classification performance, the F1-Score was used. As the harmonic mean of recall and precision, it biases neither. Further, true negative observations (which are much more numerous than the true positive observations) are not used in its calculation. The F1-Score ($F_{1}$) was calculated for each word/class/group as: $$\begin{array}{ccc} F_{1} & = & 2RP/(R+P). \end{array}$$

To understand the statistical likelihood of the calculated F1-Scores, Monte Carlo Simulation was used to estimate the significance threshold over random sampling of the confusion matrices at α=0.05 over one million simulations in accordance with accepted approaches [24]. F1-Scores of measurements higher than these thresholds were deemed to be significant. In the case of class/movement classification, these thresholds were different per class due to the uneven membership of the 19 words into the five classes of movement.

## 3. Results

Table 5 shows the significance of each measurement’s displacement (Dis.), velocity (Vel.), and acceleration (Acc.) as the average F1-Score taken over all words for each of the three participant cohorts. Values in bold denote measurements that are significant at α=0.05.

### 3.1. Differences Between Participant Cohorts

Table 6 shows absolute differences in F1-Scores between the participant cohorts with significant differences highlighted in bold. Differences in measurement classification performance between the younger and older children were seen in “Mouth Opening” (displacement and velocity), and in “Medial 1/3 Upper Action (Y)” (displacement only). Between the adults and the younger children, differences were seen in the displacement measurements of “Mouth Opening”, “Lip Action (Y)”, and “Medial 1/3 Lower Action (Y)”, together with differences across all three derivatives of “Lip Action (Z)”. Between the adults and the older children, differences were seen in the displacement and velocity of “Mouth Opening” and “Medial 1/3 Lower Action (Y)”. Differences were also seen in the displacement and acceleration of “Lip Action (Y/Z)” as well as velocity in “Lip Action (Z)”.

#### Classifying Similarity of Movement

The shaded cells in Table 7 show which measurements (rows) were expected to be significant in classifying which classes of movement (columns) based on each class’s common facial movement characteristics as given in Table 4. Only values that were found to be significant are displayed, and these are taken as the average F1-score over displacement, velocity, and acceleration for the respective measurement over all three participant cohorts.

Table 8 summarizes Table 7 into classification statistics. The scores in the expected cells of Table 7 are counted as true-positives, i.e., significant results that match expectations. Scores in the non-shaded cells are counted as false positives, i.e., significant results that do not match expectations. Shaded cells without a score are counted as false negatives, i.e., expected significance for the measurement was not observed. True negatives are ignored, i.e., expected non-significance of the measurement. The per-class significant F1-Score threshold at α=0.05 was estimated via Monte Carlo simulation using the non-uniform distribution of the 19 words (without “Papa”) into the five classes and 2432 observations (128 total participants x 19 words). For all five classes of movement individually, the measurements designed to capture the expected movements were found to be significantly able to classify the words within those movement classes. This shows that the extracted measurements capture the kinds of facial movements expected in the spoken production of the words.

Across all three cohorts the measurements found to be individually capable of classifying the words at the required level of performance (α=0.05) were as follows: “Mouth Height”, “Mouth Opening”, “Lip Action (Y/Z)”, “Medial 1/3 Action (Y)”, “Medial 1/3 Upper and Lower Action (Y/Z)”, “Pogonion (Y/Z)”, “Gonion (Y)”, “Lower Lip from Pogonion (Y)”, and “Mandibular Angle”. Confirming the classification performance of these measurements, their F1-Scores also passed the higher F1-Score thresholds estimated via Monte Carlo simulation at the α=0.01 significance level.

## 4. Discussion

In this paper, we report on the use of a state-of-the-art facial mesh detection and tracking algorithm to automatically extract and record expertly identified measurements of clinically salient facial movements from high-definition video, for the purpose of correctly classifying words from the mandibular and labial-facial stages of the MSH-PW list. Five hypotheses were evaluated and they are discussed here in turn.

### 4.1. Measurements of Anterior/Posterior Facial Movement Will Show Significant Classification Performance

In support of our first hypothesis, measurements of facial movement in depth (i.e., along the Z-axis) inferred by the detection scheme showed significant classification performance demonstrating the utility of the Blazeface detector for estimating depth from 2D video supporting. Velocity and acceleration in depth was found to perform better than displacement as evidenced by several measurements in Table 5 including “Medial 1/3 Upper/Lower Action” and “Pogonion”. This could be because whilst per frame estimations of depth may be poor, relative changes in depth across adjacent frames may be more accurate and so able to provide useful information about facial movements in that axis.

### 4.2. The Measurements with the Best Classification Performance Will Be the Same Across Participant Cohorts

Supporting our second hypothesis, Table 5 shows seven measurements with the best classification performance across all three participant cohorts. These were as follows: “Mouth Opening”, “Lip Action (Y/Z)”, “Medial 1/3 Upper Action (Y/Z)”, “Medial 1/3 Lower Action (Y/Z)”, “Pogonion (Y/Z)”, “Lower Lip from Pogonion (Y/Z)” and “Labial Fissure Width”. This suggests that assessing specific speech movement characteristics is not only feasible but also robust across different age groups. This finding provides analytical validation in the development of fit-for-purpose digital health technologies [17]. The small differences in classification performance noted across the three age cohorts in these measurements are not surprising but are worthy of further investigation. Classification performance within the adult cohort was generally much higher than within the child cohorts. The literature has clearly documented the presence of nonlinear age-related changes in speech-motor control. For example, periods of change in language and cognition can result in periods of development or regression in speech-motor performance [25]. Further, the developmental course differs not only among the different vocal tract structures [26,27,28] but also across stimuli [29] and spatial-temporal parameters [30].

### 4.3. The Measurements Designed to Capture the Distinctive Orofacial Movements Characteristic of the Words Will Perform Best in Classification

Our analysis of the results as summarized in Table 8 supports this hypothesis. This indicates the potential utility of these objective measures in supporting S-LPs in their subjective observations. In assessing motor speech control using the MSH-PW, S-LPs observe and assess specific speech movements associated with each stage. For example, stage III (mandibular control) requires the assessment of jaw range and control/stability of spoken words containing mid and low vowels as well as diphthongs. The displacement measurement of “Mouth Opening” could be used to provide an objective assessment of jaw range whereas its velocity could support the assessment of jaw control. The lateral motion of the pogonion (along the X-axis) could be used as an objective indicator of jaw stability. The preliminary data from this study are being used to inform clinical validation work this study team is currently undertaking.

### 4.4. Measurements of Lateral Deviation or Asymmetry Will Not Demonstrate Significant Classification Performance

Supporting our fourth hypothesis, as evidenced by their insignificant F1-Scores shown in Table 5 both the symmetry measurements (“Medial 1/3 Symmetry X/Y/Z”, and “Mouth Area Symmetry”) along with all of the X-axis measurements behaved according to their design in not falsely identifying any incidence of lateral deviation or asymmetry common to the (typical) production of the 20 words. Jaw deviance in the coronal plane (i.e., lateral jaw slide) has been observed in children with speech sound disorders relative to typically developing children [19,31]. Further research will be required to provide evidence that these measurements may act as objective indicators of such atypical facial movements.

### 4.5. The “Face Size” Measurement Will Not by Itself Demonstrate Significant Classification Performance

Finally, we hypothesized that the “Face Size” measurement would not exhibit classification performance above the required significance threshold because it was designed to be invariant to motion in the lower part of the face. This was borne out in our analysis; in Table 5 the “Face Size” F1-Score was found to stay under the significance threshold across all cohorts.

## 5. Conclusions

This study investigated the efficacy of facial measurements extracted from video for classifying words spoken by very young children and adults. The focus was to evaluate the ability of the facial detection scheme to extract facial measurements from video recordings that correspond with the facial movements expected by S-LPs when scoring the MSH-PW. The overall aim of this study was to show that state-of-the-art facial detection from simple video recordings can potentially be used to support S-LPs in the clinical evaluation of speech production, especially in young children.

In this study, 29 facial measurements were individually empirically validated and a common subset of these were found to be reliable discriminators of the 20 words spoken by the adult and child participants. Further, the measurements that were found to be reliable discriminators of similar movements were correlated in four out of five cases with clinical expectations of the nature of those movements [2]. As expected, measurements of lateral movement or asymmetry proved to be ineffective word classifiers due to the makeup of the participant cohorts being entirely typically developed for age. The utility of the facial detection scheme for assessing anterior/posterior facial movements was supported due to the ability of several depth (Z) measurements to classify the target wordset at a significant level of chance.

The clear results of this study show that it is feasible to extract facial measurements from 2D video recordings of participants speaking prescribed words and using the kinematic traces of these measurements to discriminate between different types of facial movement in three dimensions. This provides support for the possible clinical utility of such measurements in discriminating between productions of the same word involving typical and atypical speech motor control.

## 6. Limitations

Spoken word durations were made comparable in this work by normalizing both spatially and temporally. Spatial normalization was performed by calculating a metric of the participant’s “face size” at each frame that was designed to be invariant to speech-related movements of the lips and jaw. This allowed distance measurements of each participant’s face to be encoded as ratios of the participant’s face size. This approach, however, does not account for morphological differences in facial features between participant’s (e.g., one participant having a wider mouth relative to their face size) and so the measurements obtained are still not directly comparable without further adjustment. In addition, the extraction of such ratio-based measurements means that they cannot be easily compared to simple distance measures encoded in real world units.

The temporal normalization performed to stretch or shrink the duration of each spoken word was linear in nature and did not account for changes in speaking rate within the production of a single word by a participant. For the two child datasets, this issue was mitigated by making several recordings of a word per child and using the production with the least variance in speaking rate. A more robust approach would be to use non-linear normalization of the duration but to properly achieve this would entail identification of within word phoneme boundaries which is not a trivial problem.

This work focused only on the visually apparent component of speech production. Whilst it is useful to separate the analysis of the qualitatively different visual and audio data streams, the auditory component is at least as meaningful to an S-LP in assessing speech production abilities. As such, these data should be included in any future fully automated approach to assisting S-LPs in their assessments [9].

## 7. Future Work

Differences in motor control between different words—even for similar words such as “Map” and “Ham”—are much greater than the differences that may be present between typical and atypical productions of the same word. The classification performance of the extracted measurements demonstrated herein should not be interpreted as showing evidence for their utility in differentiating between typical and atypical speech production and further research will be conducted to investigate this use case.

This study investigated only single variable (measurement) classification performance. For use in scoring general speech production ability these measurements should be combined into a multi-variate regression framework modeled to score word production performance akin to existing paper-based approaches used by S-LPs. This would also allow for a closer examination of the contribution of each measurement (motor control characteristic) in correct speech production [9].

Manual identification of word onsets and offsets in the recorded video carries a significant time and cost burden and is a potential source of error since it can result in too much or too little lead in/out being incorporated into the kinematic traces thereby increasing the overall difference of a single measurement trace compared to the associated statistical profile. Efforts should be directed toward automating this task to increase reliability and so that it is based upon stronger objective criteria, although this may not be straightforward especially in cases of atypical speech production.

## Figures and Tables

**Figure 1 sensors-24-07235-f001:**
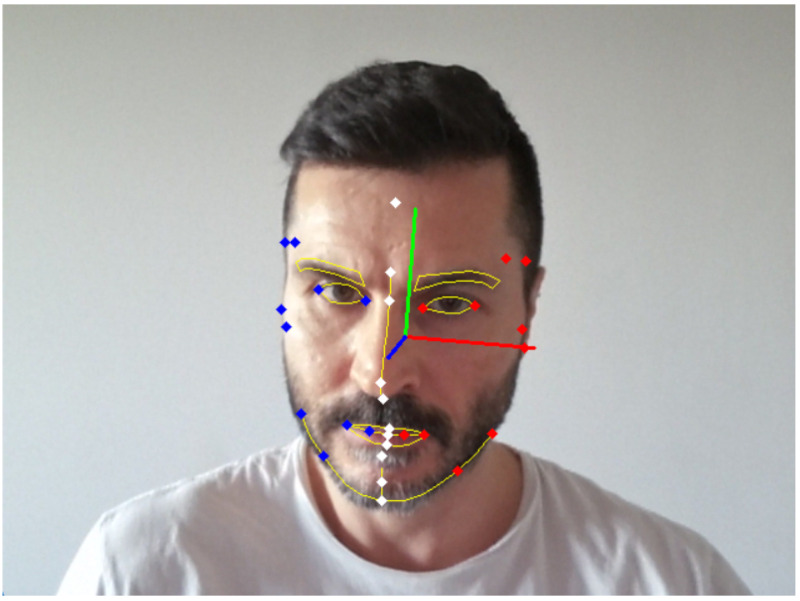
Example detection of facial landmarks and head orientation. Bilateral landmarks are shown as red and blue dots with the midline landmarks as white dots. The red, green, and blue lines depict respectively the X, Y, and Z facial axes. The main facial features are outlined in yellow.

**Figure 2 sensors-24-07235-f002:**
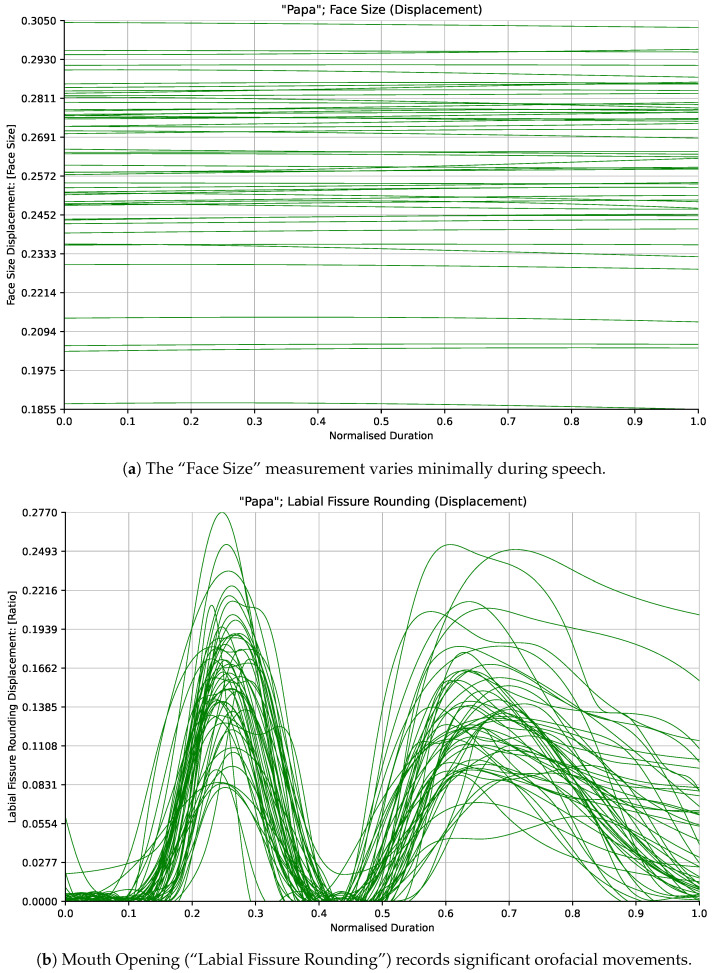
Kinematic traces from the adult participants saying the word “Pam” for the “Face Size” and “Labial Fissure Rounding” (Mouth Opening) measurements. The “Face Size” measurement stays minimally variant as the participants speak while the “Labial Fissure Rounding” (Mouth Opening) measurement captures salient orofacial movements.

**Figure 3 sensors-24-07235-f003:**
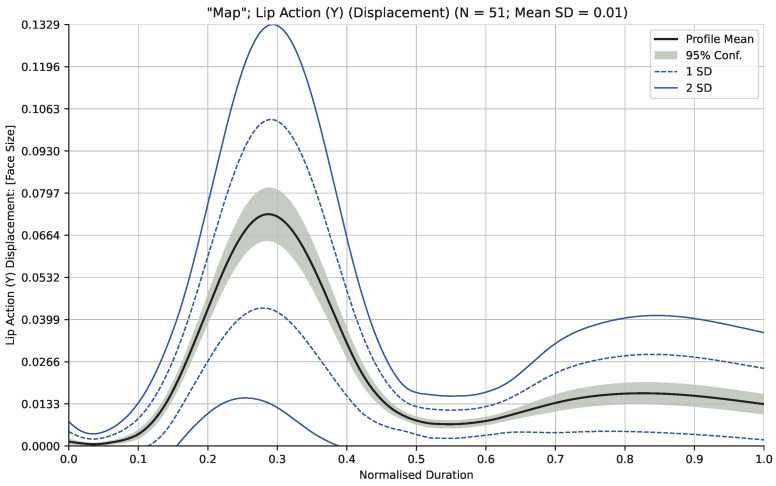
Example spatiotemporal profile of measurement “Lip Action (Y)” generated from the adult cohort for the word “Map”. Profile mean (black line), 95% confidence interval (shaded region), and 1 and 2 standard deviations (dotted and solid blue lines) shown.

**Table 1 sensors-24-07235-t001:** Child Speech and Language Characteristics.

	Younger Children Age 3.0–3.5 Years	Older Children Age 3.5–4.0 Years
Age in months (Mean; SD)	38.14 (1.84)	44.73 (1.67)
CELF-P2 Core Language SS (PR)	109.39 (69.91)	110.77 (74.26)
GFTA-3 SS (PR)	106.94 (66.06)	107.21 (66.62)
GFTA-3 Percent Phonemes Correct	89.62	92.84

Note: SS = Standard Score; PR = Percentile Rank.

**Table 2 sensors-24-07235-t002:** Association between Blazeface Mesh IDs, standard clinical facial landmarks, and the extracted measurements (Blazeface IDs in parentheses indicate average position).

Blazeface Mesh IDs	Clinical Landmarks	Measurement
199	Pogonion	Pogonion
17, 199	Labrale Inferius, Pogonion	Lower Lip from Pogonion
178, 402, 81, 311	Mid-Labial Fissure	Lip Action
178, 402, 81, 311	Mid-Labial Fissure	Medial 1/3 Symmetry
178, 402, 81, 311, 61, 291	Mid-Labial Fissure, Cheilion	Medial 1/3 Action
178, 402, 61, 291	Mid-Labial Fissure, Cheilion	Medial 1/3 Lower Action
81, 311, 61, 291	Mid-Labial Fissure, Cheilion	Medial 1/3 Upper Action
178, 402, 81, 311, 61, 291	Mid-Labial Fissure, Cheilion	Mouth Opening
0, 17, 61, 291	Labrale, Cheilion	Mouth Area Symmetry
61, 291	Cheilion	Labial Fissure Width
13, 14	Stomion	Mouth Height
172, 397, 2	Gonion, Subnasale	Gonion
199, 172, 397	Pogonion, Gonion	Mandibular Angle
93, 323, 227, 447, 168, 9, 133, 362, (68, 71), (301, 298)	Tragion, Zygion, Sellion, Glabella, Inner-Canthus, Frontotemporale	Face Size

**Table 3 sensors-24-07235-t003:** Definitions of the measurements. Scalars are suffixed with ^+^ and a * suffix indicates that the measurement was not normalized using the “Face Size” factor.

Measurement	Definition	Physiology
Pogonion	Displacement of the chin tip (pogonion).	Mandibular range/control.
Lower Lip from Pogonion	Displacement of the inferior lip midline (stomion inferius) relative to pogonion.	Appropriate labial fissure close/open phase.
Lip Action	Displacement of the mean inferior medial 1/3 (of the labial fissure) with respect to the superior.	Appropriate labial fissure close/open phase.
Medial 1/3 Symmetry	Difference in the participant’s left half of the medial 1/3 labial fissure with the right half.	Labial symmetry.
Medial 1/3 Action	Displacement of the medial 1/3 labial fissure with respect to the mouth corners (cheilion).	Independent bilabial control and appropriate labial fissure rounding/retraction.
Medial 1/3 Lower Action	Displacement of the inferior medial 1/3 labial fissure with respect to the mouth corners (cheilion).	Inferior labial control.
Medial 1/3 Upper Action	Displacement of the superior medial 1/3 labial fissure with respect to cheilion.	Superior labial control.
Mouth Opening *^+^	Ratio of vertical inferior and superior lip distance at midline with labial fissure width.	Appropriate mandibular range/control.
Mouth Area Symmetry ^+^	Difference in area of the left half of the labial fissure with the right. Half labial fissure area is that of a triangle with corners stomion superior/inferior and that half’s cheilion.	Labial fissure symmetry.
Labial Fissure Width ^+^	Absolute lateral distance between cheilion.	Appropriate labial fissure rounding/retraction.
Mouth Height ^+^	Vertical distance from the facial origin to the midpoint position of stomion superius and stomion inferius.	Appropriate labial fissure close/open phase.
Gonion	Displacement of the mean of the most posterior and inferior points of the mandible (gonion).	Mandibular range/control.
Mandibular Angle *^+^	The angle a line from pogonion and the bilateral gonial landmark mean makes with the anterior/posterior facial orientation about the transverse axis.	Mandibular range/control.
Face Size	A multidimensional measurement derived from several distances around the forehead and eyes designed to be invariant during participant speech motor movements. Used to normalize other measurements to facilitate inter-participant comparisons.	N/A

**Table 4 sensors-24-07235-t004:** Mandibular and labial-facial movement characteristics.

Class	Words	MSH-PW Level	Lip Shape	Distinguishing Characteristics
1	Ba, Map, Ham, Pam	Mandibular	Vertical	Low Vowel Height.
2	Eye, Pie	Mandibular	Vertical	Diphthongs.
3	Umm, Bob, Pup	Mandibular	Vertical	Mid Vowel Height.
4	Bee, Peep, Feet	Labial Facial	Horizontal	Lip action involves spreading (retraction).
5	Boy, Bush, Moon	Labial Facial	Horizontal	Lip action involves rounding, syllable contains bilabials.
5	Phone, Fish, Wash, Show	Labial Facial	Horizontal	Lip action involves rounding, syllable contains fricatives.

**Table 5 sensors-24-07235-t005:** Measurement classification average F1-Scores by cohort; significant values in bold.

	Adult			Older Children			Younger Children		
Measurement Thresholds @ α=0.05	Dis. 11.0	Vel. 11.0	Acc. 11.0	Dis. 10.9	Vel. 10.9	Acc. 10.9	Dis. 11.5	Vel. 11.5	Acc. 11.5
Labial Fissure Width	**11.3**	**15.8**	**15.3**	**12.1**	**13.6**	**12.9**	9.3	**13.8**	**13.9**
Mouth Height	6.7	9.9	**11.5**	9.8	**12.9**	**13.3**	8.5	7.4	11.3
Medial 1/3 Symmetry (X)	4.4	4.6	5.2	4.8	4.4	4.3	3.9	4.4	4.5
Medial 1/3 Symmetry (Y)	4.9	5.1	4.2	6.5	7.1	7.8	5.8	5.3	7.0
Medial 1/3 Symmetry (Z)	4.5	5.8	5.7	4.8	6.0	6.5	4.6	4.6	5.2
Mouth Area Symmetry	4.9	4.4	5.0	4.6	4.6	4.8	3.7	4.6	4.9
Mouth Opening	**49.8**	**44.5**	**26.6**	**39.6**	**31.7**	**20.9**	**37.2**	**34.8**	**23.8**
Lip Action (X)	7.0	6.7	6.6	6.3	4.5	6.1	4.8	5.3	6.4
Lip Action (Y)	**42.4**	**38.9**	**34.0**	**35.9**	**34.1**	**28.1**	**30.9**	**30.8**	**22.7**
Lip Action (Z)	**41.9**	**39.8**	**33.2**	**28.2**	**29.4**	**23.9**	**25.1**	**28.8**	**21.8**
Medial 1/3 Action (X)	4.1	4.2	4.4	4.3	5.1	5.7	4.6	4.3	5.3
Medial 1/3 Action (Y)	7.3	9.6	10.6	10.1	**12.0**	**12.8**	8.6	9.7	9.6
Medial 1/3 Action (Z)	7.2	7.9	8.7	7.5	8.6	8.5	5.5	7.9	6.9
Medial 1/3 Upper Action (X)	4.1	4.9	5.0	4.3	5.5	5.2	5.3	5.6	6.3
Medial 1/3 Upper Action (Y)	**27.1**	**33.7**	**27.9**	**31.4**	**32.1**	**24.2**	**21.5**	**28.2**	**23.4**
Medial 1/3 Upper Action (Z)	**11.6**	**14.4**	**15.9**	**15.4**	**17.0**	**17.4**	**12.7**	**15.6**	**15.0**
Medial 1/3 Lower Action (X)	4.4	4.2	3.9	4.5	4.7	5.0	4.6	4.3	4.8
Medial 1/3 Lower Action (Y)	**37.4**	**37.6**	**29.3**	**29.5**	**29.2**	**21.8**	**26.6**	**29.3**	**20.8**
Medial 1/3 Lower Action (Z)	9.7	**11.5**	**11.5**	**13.7**	**12.2**	**15.3**	10.2	**12.1**	**12.1**
Pogonion (X)	4.6	4.7	5.1	5.7	7.0	6.7	4.7	4.7	5.3
Pogonion (Y)	7.2	10.5	**12.4**	**11.0**	**17.2**	**17.0**	8.6	10.3	**14.1**
Pogonion (Z)	**12.2**	**19.1**	**19.3**	6.9	**22.5**	**21.1**	3.8	**19.3**	**17.1**
Gonion (X)	5.2	4.6	4.8	4.7	5.5	5.2	6.1	4.0	4.5
Gonion (Y)	5.0	5.6	6.3	5.8	6.6	7.5	4.5	7.3	6.6
Gonion (Z)	6.2	7.5	8.7	5.3	6.4	8.2	4.3	7.8	7.2
Lower Lip from Pogonion (X)	4.7	5.1	4.8	5.1	5.9	5.8	5.3	4.8	5.4
Lower Lip from Pogonion (Y)	**14.2**	**23.4**	**20.6**	**17.1**	**24.8**	**22.2**	**13.0**	**16.6**	**16.4**
Lower Lip from Pogonion (Z)	5.4	6.4	5.7	4.9	6.7	5.8	4.0	4.8	5.0
Mandibular Angle	6.7	9.0	10.0	**11.1**	**12.4**	**12.0**	8.9	**15.1**	**12.1**
Face Size	4.0	6.4	7.3	5.4	8.7	8.1	5.2	7.9	6.9

**Table 6 sensors-24-07235-t006:** Measurement classification F1-Score differences between cohorts; significant values in bold.

	Adults vs. Older Children			Adults vs. Younger Children			Older Children vs. Younger Children		
Measurement Thresholds @ α=0.05	Dis. 11.0	Vel. 11.0	Acc. 11.0	Dis. 11.2	Vel. 11.2	Acc. 11.2	Dis. 11.2	Vel. 11.2	Acc. 11.2
Labial Fissure Width	4.5	5.0	8.1	4.5	5.2	7.9	5.6	3.6	6.0
Mouth Height	6.2	6.0	6.3	4.1	5.5	6.5	5.8	8.3	5.7
Medial 1/3 Symmetry (X)	2.9	3.7	3.4	3.2	2.9	3.9	2.6	3.6	3.9
Medial 1/3 Symmetry (Y)	4.2	4.3	5.4	3.8	4.5	4.9	3.8	5.8	6.1
Medial 1/3 Symmetry (Z)	3.2	4.6	3.2	2.8	3.7	3.4	3.3	5.5	4.7
Mouth Area Symmetry	2.3	3.3	3.0	3.1	3.8	3.3	2.7	3.1	4.3
Mouth Opening	**17.0**	16.8	9.5	**14.5**	**11.4**	8.9	**13.3**	**14.5**	**12.3**
Lip Action (X)	4.3	4.1	5.2	4.4	4.7	4.7	3.4	4.3	4.7
Lip Action (Y)	**15.2**	**11.1**	**14.3**	**13.4**	10.6	**11.9**	10.2	8.8	9.8
Lip Action (Z)	**17.8**	**13.8**	**12.2**	**16.8**	**11.5**	**13.1**	10.2	10.0	7.9
Medial 1/3 Action (X)	3.0	3.9	2.8	3.0	2.6	2.9	2.6	3.9	1.8
Medial 1/3 Action (Y)	4.3	5.2	5.6	4.2	4.7	5.9	4.9	5.4	6.7
Medial 1/3 Action (Z)	3.3	5.6	5.7	3.8	3.3	4.6	2.9	5.5	5.0
Medial 1/3 Upper Action (X)	3.2	4.5	2.5	4.6	3.7	3.5	3.9	4.5	3.5
Medial 1/3 Upper Action (Y)	9.3	10.0	9.3	8.1	9.6	8.2	10.9	**11.2**	9.0
Medial 1/3 Upper Action (Z)	6.6	7.5	7.9	5.4	7.0	7.3	6.3	4.8	4.9
Medial 1/3 Lower Action (X)	2.9	3.5	3.1	2.3	2.9	3.7	2.9	2.7	2.9
Medial 1/3 Lower Action (Y)	**14.1**	**13.2**	11.1	**11.6**	**11.9**	11.2	**11.4**	**12.2**	7.7
Medial 1/3 Lower Action (Z)	6.3	7.4	8.1	3.8	4.3	6.4	5.1	6.2	7.5
Pogonion (X)	3.1	4.1	4.1	2.6	2.3	3.7	3.4	4.2	5.3
Pogonion (Y)	5.0	9.2	8.6	3.5	5.4	5.7	5.3	9.4	7.1
Pogonion (Z)	5.6	7.6	7.6	8.4	7.3	7.6	5.3	7.5	6.7
Gonion (X)	2.3	2.9	3.1	2.9	3.4	3.0	3.4	4.5	5.3
Gonion (Y)	2.8	4.0	4.4	2.6	3.5	3.0	2.5	5.1	4.8
Gonion (Z)	3.2	3.4	4.5	3.1	5.2	5.2	3.1	4.4	4.0
Lower Lip from Pogonion (X)	2.5	3.6	4.2	3.1	3.5	3.2	2.5	4.0	4.2
Lower Lip from Pogonion (Y)	7.0	7.7	8.5	4.5	9.0	6.0	8.2	9.5	9.3
Lower Lip from Pogonion (Z)	2.8	5.1	4.2	3.0	4.8	4.3	3.1	4.2	4.9
Mandibular Angle	6.0	6.2	7.5	3.7	7.5	5.4	5.8	6.8	6.7
Face Size	2.9	4.6	4.6	3.8	3.3	3.8	3.0	4.3	5.1

**Table 7 sensors-24-07235-t007:** Similar facial movement significance across all cohorts. The shaded cells show measurements associated with facial movements that were expected to be present for that class of movement.

Class	1	2	3	4	5
No. Words	4	2	3	3	7
Labial Fissure Width	40.1			28.7	52.2
Mouth Height	35.3	19.9		24.1	
Medial 1/3 Symmetry (X)					
Medial 1/3 Symmetry (Y)					
Medial 1/3 Symmetry (Z)	25.1				
Mouth Area Symmetry					
Mouth Opening	54.4	41.9	56.8	31.4	64.3
Lip Action (X)					
Lip Action (Y)	58.8	42.0	51.3	27.6	70.0
Lip Action (Z)	48.9	35.0	53.4	28.5	59.2
Medial 1/3 Action (X)					
Medial 1/3 Action (Y)	41.4				49.4
Medial 1/3 Action (Z)					
Medial 1/3 Upper Action (X)					
Medial 1/3 Upper Action (Y)	43.4	32.1	51.7	27.9	59.0
Medial 1/3 Upper Action (Z)	40.8	30.2	22.8		50.1
Medial 1/3 Lower Action (X)					
Medial 1/3 Lower Action (Y)	54.4	35.1	49.2	29.1	63.7
Medial 1/3 Lower Action (Z)	33.4	29.3		27.2	
Pogonion (X)					
Pogonion (Y)	35.5	21.6	23.5	26.2	
Pogonion (Z)	39.6	20.5	33.2		52.7
Gonion (X)					
Gonion (Y)	27.1				
Gonion (Z)					
Lower Lip from Pogonion (X)					
Lower Lip from Pogonion (Y)	45.9	22.7	29.1		54.8
Lower Lip from Pogonion (Z)					
Mandibular Angle	33.5	27.7		24.1	
Face Size					

**Table 8 sensors-24-07235-t008:** Similar facial movement statistics and classification significance by extracted measurements.

Class	1	2	3	4	5	ALL
TP	6	3	2	4	2	17
FP	10	9	7	6	8	40
FN	0	1	1	1	1	4
Recall	1.00	0.75	0.67	0.80	0.67	0.81
Precision	0.38	0.25	0.22	0.40	0.20	0.30
F1-Score	0.55	0.38	0.33	0.53	0.31	0.44
F1-Score Significance Threshold	0.23	0.16	0.21	0.20	0.29	0.21

## Data Availability

The data presented in this study are available on request from the corresponding author due to participant privacy restrictions.

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
