# Peer review of "Facial Movements Extracted from Video for the Kinematic Classification of Speech"

_sensors, 2024, doi:10.3390/s24227235_

Round 1
Reviewer 1 Report
Comments and Suggestions for Authors
The introduction is clear and the goal is well defined.
The description of the proposed method must be improved.
It's not clear how 2d landmark coordinates are mapped into 3d space.
An additional figure should be provided alongside Table 3 to better understand what the masurments rapresent.
The classication algorithm description should be improved by adding equations to explain better how the classification is performed.
Author Response
The introduction is clear and the goal is well defined.
Thank you.
The description of the proposed method must be improved.
It's not clear how 2d landmark coordinates are mapped into 3d space.
Thank you for this comment. We have included text at line 193 before Table 2 in section 2.3.1 explaining that the method by which the Blazeface detector maps 2D pixels to 3D points is outside the scope of this study and the reader is advised to refer to the cited source material on the subject.
An additional figure should be provided alongside Table 3 to better understand what the masurments rapresent.
Thank you for this suggestion. We originally considered displaying the measurements annotated on the face. However, we felt this would be of limited value since the nature of the measurements largely relates to proportional degrees of travel in fixed 3D directions which would be difficult to show adequately superimposed on a 2D facial schematic or photo. Additionally, the measurements cluster around the mouth and jaw and it was felt that their close proximity would likely be too confusing for the reader. As a result we settled on a tabulated description of the measurements, and a cross-reference of the associated landmarks to the underlying facial mesh (as indicated in Table 2). We believe this combination of information allows for a clear and detailed understanding of the nature of the facial measurements sufficient for the work to be replicated.
The classication algorithm description should be improved by adding equations to explain better how the classification is performed.
Thank you for this suggestion. In addition to our description of the procedure starting at line 260 (section 2.3.2 on page 10), we have now added an equation showing the calculation of the similarity metric which is used as the classifying function. In addition, we have now added an equation that explicitly shows how we select the predicted word.
Reviewer 2 Report
Comments and Suggestions for Authors
This paper proves that it is feasible to extract the kinematic classification of speech based on facial movements features, and the method is novel, but it is suggested to make some revisions. (1)As mentioned in the article, since the subjects occasionally exist common obstructions included hand(s) in front of the face, a bowed head, or head turned away, what is the effectiveness of the method when there are different occlusion degrees? No explanation has been given in this paper. (2) If the importance of different areas can be presented in combination with facial heatmap, the effect will be better displayed.
Author Response
As mentioned in the article, since the subjects occasionally exist common obstructions included hand(s) in front of the face, a bowed head, or head turned away, what is the effectiveness of the method when there are different occlusion degrees? No explanation has been given in this paper.
Thank you for this comment. It is unclear at this time how effective the method is in the presence of different kinds of occlusions, and since evaluating the effectiveness of the method in such situations was not within the scope of this work, we purposefully designed the procedures around our data collection to allow for multiple repetitions of a word by the (child) participants until a sufficiently acceptable production was produced with no occlusions. As stated in our procedures at line 165, the best production was retained.
If the importance of different areas can be presented in combination with facial heatmap, the effect will be better displayed.
Thank you for this suggestion. We did consider this kind of visualisation originally. However, because the extracted measurements are typically derived between landmark pairs (e.g., as distances), a heatmap style of representing them would be misleading due to larger regions of the facial surface being indicated than are actually involved in the calculation of the measurements.